# Comparison of the Outcomes between Systematic Lymph Node Dissection and Lobe-Specific Lymph Node Dissection for Stage I Non-small Cell Lung Cancer

**DOI:** 10.3390/diagnostics13081399

**Published:** 2023-04-12

**Authors:** Ching-Chun Huang, En-Kuei Tang, Chih-Wen Shu, Yi-Ping Chou, Yih-Gang Goan, Yen-Chiang Tseng

**Affiliations:** 1Division of Thoracic Surgery, Department of Surgery, Kaohsiung Veterans General Hospital, Kaohsiung 813, Taiwan; 2School of Medicine, National Yang Ming Chiao Tung University, Taipei 112, Taiwan; 3Institute of BioPharmaceutical Sciences, National Sun Yat-Sen University, Kaohsiung 804, Taiwan; 4Department of Biomedical Science and Environmental Biology, Kaohsiung Medical University, Kaohsiung 807, Taiwan; 5Division of Trauma, Department of Emergency, Kaohsiung Veterans General Hospital, Kaohsiung 813, Taiwan; 6Department of Surgery, Kaohsiung Veterans General Hospital Pingtung Branch, Pingtung 900, Taiwan; 7Institute of Clinical Medicine, National Yang Ming Chiao Tung University, Taipei 112, Taiwan

**Keywords:** L-SND, NSCLC

## Abstract

Background: This study compares the surgical and long-term outcomes, including disease-free survival (DFS), overall survival (OS), and cancer-specific survival (CSS), between lobe-specific lymph node dissection (L-SND) and systematic lymph node dissection (SND) among patients with stage I non-small cell lung cancer (NSCLC).Methods: In this retrospective study, 107 patients diagnosed with clinical stage I NSCLC undergoing video-assisted thoracic surgery lobectomy (exclusion of the right middle lobe) from January 2011 to December 2018 were enrolled. The patients were assigned to the L-SND (*n* = 28) and SND (*n* = 79) groups according to the procedure performed on them. Demographics, perioperative data, and surgical and long-term oncological outcomes were collected and compared between the L-SND and SND groups. Results: The mean follow-duration was 60.6 months. The demographic data and surgical outcomes and long-term oncological outcomes were not significantly different between the two groups. The 5-year OS of the L-SND and SND groups was 82% and 84%, respectively. The 5-year DFS of the L-SND and SND groups was 70% and 65%, respectively. The 5-year CSS of the L-SND and SND groups was 80% and 86%, respectively. All the surgical and long-term outcomes were not statistically different between the two groups. Conclusion: L-SND showed comparable surgical and oncologic outcomes with SND for clinical stage I NSCLC. L-SND could be a treatment choice for stage I NSCLC.

## 1. Introduction

As the prevalence of non-smoking-related lung cancer has gradually increased in Asian countries, a low-dose computed tomography (LDCT) screening program has been implemented, bringing about a decrease in lung cancer mortality rates and a shift in the stage distribution toward earlier stages [1]. In Asia, LDCT has further increased the incidents of persistent subsolid nodules [2]. A high rate of persistent subsolid nodules is associated with lung adenocarcinoma [3]. Therefore, the application of thoracoscopic lung sparing surgery is more extensive. How to consider the patient’s prognosis and reduce the scope of surgical resection and complications has become a major clinical factor. For the staging of lung cancer, mediastinoscopy was used to evaluate mediastinal lymph node (LN) metastasis in the past, and it was gradually replaced recently by whole-body positron emission tomography (PET), since it allows less-invasive and non-operative examinations [4].

For patients diagnosed with early-stage non-small cell lung cancer (NSCLC),the standard treatment is lobectomy accompanied by systematic lymph node dissection (SND) [5]. In recent decades, several studies showed that lobe-specific lymph node dissection (L-SND) had no influence on overall survival (OS) or disease-free survival (DFS) and could reduce the risk for perioperative complications [6,7]. Additionally, several publications reported that systematic lymph node dissection was associated with greater blood loss, longer operative time, chylothorax, recurrent nerve palsy, and greater chest tube drainage, in contrast to lymph node sampling [6,8,9,10]. Otherwise, the lymph node mapping and lymphatic drainage rout of mediastinum were established [11,12,13]. In tumors located in the upper lobe, the lymph nodes commonly spread to the upper mediastinum, whereas those located in the lower lobe spread to the lower mediastinum. Skip metastasis was still observed, but rarely.

Since video-assisted thoracic surgery (VATS) was introduced, it has been adopted as a treatment for lung cancer. VATS has also been reported to reach similar long-term outcomes in early-stage lung cancer with fewer complications [14,15,16,17]. With the abovementioned perspectives, we hypothesized that lobectomy with L-SND by VATS could be taken into consideration for those patients with early-stage NSCLC. In our clinical experience, patients always ask if thoracoscopic surgery is available or not, and they are concerned with the length of wound or chest drainage. We hope to conduct less damage during the surgery, expecting to lessen the post-operative discomfort. Thus, the present study focused on the comparison of the surgical and long-term outcomes, including DFS, OS, and cancer-specific survival (CSS), between L-SND and SND among patients with stage I NSCLC.

## 2. Materials and Methods

Data of clinical stage I NSCLC patients who underwent a lobectomy from January 2011 to December 2018 were collected from the database of Kaohsiung Veteran General Hospital. The exclusion criteria were as follows: patients who received wedge excision or segmentectomy; patients who received right middle lobe lobectomy; patients with secondary lung cancer; and those who were lost to follow-up (Figure 1). The reason we excluded patients that underwent RML lobectomy is that there was no specific lymphatic spread pattern [18]. Altogether, 107 patients were enrolled in our study.

All patients’ data on demographics, operative course, clinical and pathologic staging, and postoperative follow-up were recorded. Preoperatively, each patient was staged with a chest and brain CT, whole-body bone scan, abdominal sonography, and whole-body PET. The mediastinal lymph nodes were defined clinically negative if they were <10 mm onthe chest CT and were not hypermetabolic on the PET scan. Mediastinoscopy was not routinely used in our study. All patients underwent complete resection with negative tumor margins by VATS. The specimens were reviewed by a specialist from the pathology department.

The mediastinal lymph nodes were categorized into upper zone (stations 2–4 on the right, and stations 5 and 6 on the left) and lower zone (stations 7–9) lymph nodes [19]. The margin of station 2 (Figure 2) is the upper paratracheal nodes. Their upper border is the apex of the right lung and pleural space, and the lower border is the intersection of the caudal margin of the innominate vein with the trachea. The upper border of the mediastinum is in the midline.

Station 3 is the prevascular and retrotracheal nodes, which are located from the apex of the chest to the level of the carina. The anterior border of station 3 is the posterior aspect of the sternum, and the posterior border is the anterior border of the superior vena cava. Station 4, the right lower paratracheal nodes, is within the intersection of the caudal margin of the innominate vein with the trachea and lower border of azygos vein. The margin of station 5 (Figure 3) is between the lower border of the aortic arch and the upper rim of the left main pulmonary artery, while station 6 is within a line tangential to the upper border of the aortic arch and the lower border of the aortic arch.

Subcarinal nodes are defined as station 7 (Figure 4), which is located from the carina of the trachea to the upper border of the lower lobe bronchus.

Beneath station 7 is station 8, whose lower border is the diaphragm. Station 9 (Figure 5) is defined bilaterally from the inferior pulmonary vein to the diaphragm.

Our patients with upper-lobe tumors received upper-zone lymph node dissection (LND), and those with lower-lobe tumors received lower-zone LND. These patients were assigned to the L-SND group, also referred to as group 1. Patients receiving both upper and lower zone LNDs, on the other hand, were assigned to the SND group, also referred to as group 2.

The criteria of selecting the L-SND group was a tumor less than 30mm, no enlarged lymph nodes detected by a preoperative chest CT or those not hypermetabolic on a PET scan, and no obvious lymphadenopathy in the non-related station of the tumor during the operation. If patients did not undergo a PET scan and lymphadenopathy was observed by a chest CT, a perioperative frozen section was performed. Patients with benign frozen reports of mediastinal LNs were categorized to the L-SND group, while the others were categorized to the SND group. All patients categorized to the L-SND group received the lymph node dissection of related stations. Detailed procedures and anatomical landmarks of lymph node dissection were as follows. In the right upper mediastinal zone (stations 2 and 4), all fat tissue with LNs between the phrenic nerve, vagus nerve, right innominate artery, and right main bronchus were removed, exposing the superior vena cava, trachea, anterolateral aspect of the ascending aorta, right tracheobronchial angle, azygos vein, and right main bronchus. In the aortopulmonary zone (stations 5 and 6), all fat tissues with LNs between the phrenic and vagus nerves were removed down to the left main pulmonary artery (PA), including the subaortic space. It should be noted that we did not divide the ligamentum arteriosum, but we did identify the left recurrent laryngeal nerve along the ligament. In the subcarinal zone (station 7), all the subcarinal tissue was removed, ex-posing the right and left main bronchi and posterior pericardium. In the lower mediastinal zone, station 8 and 9 nodes were removed by clearing all LNs around the inferior pulmonary vein, esophagus, and pulmonary ligament [20]. If patients were initially selected for the L-SND group but hardly underwent lymph node dissection, they would shift to the SND group.

### Statistical Analysis

The clinical characteristics of patients with L-SND and SND were compared using the independent sample *t*-test, chi-square test, and Kaplan–Meier test. The independent sample *t*-test was used to calculate the age, pre-operative (pre-OP) tumor size, surgical time, admission duration, intensive care unit (ICU) duration, and forced expiratory volume in 1 s (FEV1)/forced vital capacity (FVC). The chi-square test was applied for analyzing data on sex, smoking history, histological findings, tumor location, adjuvant chemotherapy, clinical stage, differentiated type or not, underlying disease, and pathologic N (pN) stage. The Kaplan–Meier method was used to analyze DFS, OS, and CSS. Admission duration was defined as the duration from the day the patient underwent the surgery to the day of discharge. DFS was defined as the time interval from the patient receiving surgery to the first diagnosis of locoregional or distant disease recurrence or until the last follow-up. For the calculation of DFS, patients who died without recurrence or who were known to have no recurrence at the date of the last follow-up were censored. Patients who died of unnatural causes were excluded from the OS analysis, but they were included in the CSS analysis.

Locoregional recurrence was defined as the presence of any recurrent disease within the ipsilateral hemithorax or mediastinum. All other sites of recurrence were referred to as distant metastases. The length of OS was defined as the interval between the date of surgical intervention and death due to any cause or the last follow-up. To avoid calculating survival from a small number of observations, the data for DFS and OS curves were censored at 60 months for patients without recurrence or mortality. The IBM SPSS statistics version 20 was used for all statistical analyses. A two-tailed *p*-value of 0.05 was considered significant.

## 3. Results

### 3.1. The Study Cohort

The study cohort comprised 107 patients with clinical stage I NSCLC who received VATS lobectomy. In total, 28 and 79patients were assigned to the L-SND and SND groups, respectively. Totally, 72.9% of patients received a PET scan preoperatively. A total of 71.4% patients in group 1 and 73.4% in group 2 received a PET scan individually. There were no statistical differences in the baseline characteristics (Table 1), such as age, sex, smoking history, tumor histology, location, and size, adjuvant chemotherapy, clinical stage, differentiated type or not, pathological stage, underlying disease, and FEV1/FVC ratio between the two groups. There was a datum of FEV1/FVC lost.

### 3.2. Surgical Outcome

No patient had conversion to thoracostomy from VATS. Two patients developed postoperative complications. One had a hemothorax after the surgery, suspecting surgical wound hemorrhage in group 2. The symptoms were relieved after blood transfusion and drainage by using a chest tube. The other patient had postoperative chylothorax and persistent air leakage in group 1. The patient recovered after conservative management but had prolonged hospitalization of up to 37 days. All the patients did not undergo a second operation. Altogether, one patient’s data on surgical time along with 11 patients’ data on ICU durationwere lost. There were no statistical differences in surgical time, admission duration, and ICU duration (Table 2).

### 3.3. Survival Outcomes

The mean follow-up duration was 60.6 months. The 5-year OS rate was 80%, while the mean DFS rate was 74%. There were no statistical differences in OS (*p* = 0.566) (Figure 6), DFS (*p* = 0.497) (Figure 7), and CSS (*p* = 0.813) (Figure 8) between the two groups.

The 5- and 10-year OS rates of the L-SND group were 82% and 63%, respectively, whereas those of the SND group were 84% and 71%, respectively. The 5- and 10-year DFS rates of the L-SND group were 70% and 70%, whereas those of the SND group were 65% and 62%, respectively. The CSS rates of the L-SND group were 80% and 80%, whereas those of the SND group were 86% and 73%, respectively.

## 4. Discussion

Since the 1990s, it has been reported in several studies that the L-SND of NSCLC has had similar outcomes with and lesser morbidity rates than SND [14,21,22]. Okada and colleagues [14] concluded that the selective mediastinal LND of stage I NSCLC was an alternative to curative surgery, and no difference in the 5-year DFS (L-SND: 74.6%; SND: 73.4%) and OS (L-SND: 81.9%; SND: 79.7%) rates was noted. However, surgical procedures such as VATS or a thoracostomy were not mentioned and discussed. In this case, the complication rate and admission duration might be left for further research.

In a previous retrospective study, Ishiguro and colleagues [22] showed that there was no difference between complete mediastinal LND and selective dissection. They included the patients with clinical stages I to III, and the operations varied from lobectomy to lobectomy with adjacent organ resection. The complexity of the study has resulted in worse survival outcomes, as compared to our study. The previous study’s 5-year OS rates were 76% and 71.9% for the L-SND and SND groups, respectively, whereas, in our study, these were 82% and 84%, respectively. Hishida and colleagues [14] performed a multi-institutional retrospective study with a propensity score analysis, in which they compared the outcomes between L-SND and SND among patients with stage I and II NSCLCs. The OS rate was not significantly different between the two groups (L-SND: 81.5%; SND: 75.9%), but there was a lack of peri-operative analysis, such as the surgical time.

Scott and colleagues [23] performed a secondary analysis of data from the American College of Surgeons Oncology Group Z0030 randomized clinical trial that compared VATS with an open lobectomy for lung cancer. Although bilobectomy, lobectomy, and segmentectomy are included in the procedures, the VATS group demonstrated a shortened operative time and hospital length of stay. The rates of at least one complication and chest tube drainage for <7 days were confirmed lower in the VATS group. Lee and colleagues^10^ conducted a retrospective review of a prospective database under a propensity score-matched analysis comparing the long-term outcomes of NSCLC treated by VATS with those bythoracotomy. During the 36 months of mean follow-up, the OS and DFS rates were not different statistically. The 5-year OS rates were 74.9% and 76.6% in VATS and thoracostomy groups, respectively (*p* = 0.767). They reported a decreased 5-year DFS rate in the VATS group, but this was not statistically significant (*p* = 0.552). Overall, VATS was chosen as the surgical procedure in our study because it could lessen the complications and retain similar survival.

Lobectomy with LND was performed as the surgical procedure in our study, and lobectomy mainly occupied the surgical time. Scott and colleagues’ secondary analysis [23] showed that the operative time of lobectomy via thoracostomy was longer than that of VATS (mean, 117.5 min for VATS vs. 171.5 min for thoracostomy; *p* < 0.001). To reduce the bias, we performed VATS for all patients in our studies, which accounted for no statistically significant difference between the two groups. In addition, only two patients had postoperative complications. Therefore, we took the ICU and admission durations as our surgical outcomes.

Chylothoraxis an annoying complication in pulmonary resection, occurring postoperatively. It occurs owing to injury of the thoracic duct and is highly related to aggressive mediastinal lymph node dissection [24]. The incident is about 0.7% to 2.5%. Conservative treatment through chest tube drainage and by maintaining the patient’s nutritional balanceis the first choice. Severely, a second surgery is necessary if the chest drainage is greater than 1000mL per day or if there is continued drainage for more than 14 days. Both management would postpone the admission period about 7 days or more. We initially put our efforts into analyzing the duration of the chest tube remaining, but the data collection and analysis were challenging. We thus replaced the chest tube duration with admission duration because the patients were usually discharged the next day when the chest tube was removed. In this study, we aimed to discover that the patients undergoing L-SND could shorten the admission period because the less-mediastinal lymph nodes were dissected. However, the admission duration in the L-SND group seemed to be longer in our study. We speculate that this might have resulted from the fact that only one case developed a complication. It turns out that no statistical difference was observed between the two groups.

Nodal upstaging has been observed in NSCLC, which is defined as the presence of unsuspected pathologic hilar (pN1) or mediastinal (pN2) disease detected during the final histopathologic evaluation of surgical specimens. These patients initially have clinical N0 diseases [25]. According to the Cancer and Leukemia Group B prospective clinical trial (CALGB 9761) [26], by excluding other malignant disease and benign processes, only 71.6% of patients with clinical stage I NSCLC disease retained that stage and diagnosis after complete surgical staging. A total of 14% of patients upgraded to stage II disease, while 13.5% upgraded to stage III. For evaluating the risk factors of inaccuracy in N staging, relevant research has been conducted. Al-Sarraf and colleagues [27] concluded that inaccurate nodal staging was common in patients with a history of tuberculosis, rheumatoid arthritis, and diabetes mellitus, bringing about the false detection by PET scan. The PET detection of primary NSCLC and its associated lymph nodal involvement is solely based on the metabolic uptake of 2-[18F] fluoro-2-deoxy-D-glucose (FDG) by tumors that are influenced by the inflammatory process, causing the inaccurate staging. Boffa and colleagues [28] also underlined that positron emission tomography did not improve staging accuracy. A retrospective study by Park and colleagues [29] reported that nodal upstaging measured by PET was 14.3%. In our studies, the total number of people with pN1 disease was 18 (16.8%), whereas the number with pN2 was 18 (16.8%), which are similar to the early cohort, and there was no statistical difference in the two groups. Hence, in this study, L-SND was proven to be not inferior to the standard SND.

Compared to thoracotomy, VATS has been accepted in the last decades. It is beneficial in decreasing pain, in less serious wounds, in shortening the length of postoperative stay, and in increasing the compliance to adjuvant chemotherapy [23]. Although the issues of incomplete lymph node dissection or less lymph nodes samples were mentioned [30,31], it was revised recently. Samayoa and colleagues even concluded that the number of LNs removed was correlated with an improved survival [32]. D’Amico and colleagues compared 199 patients from the NCCN NSCLC Database undergoing VATS with 189 patients undergoing thoracostomy and found similar numbers of N1 + N2 stations resected [33]. However, the complication rates as well as the oncologic outcomes were not mentioned. Kneuertz and colleagues collected patients with stage I-IIIa NSCLC who had undergone lobectomy that was robotic-assisted, VATS, or the open approach at a single center [34]. The total lymph node count and upstaging rate were similar in the three groups. Kneuertz and colleagues also analyzed the oncologic outcomes, the median follow-up period of which was 44.8 months. The stage-specific survival rates were similar in the three groups, and the 5-year OS of stage Ib was 70% in their studies. There is no denying that all surgical approaches of LN dissection are feasible in the surgery of lung cancer. Stephens and colleagues specifically established a cohort to make a comparison between thoracotomy and VATS groups in clinical stage I NSCLC [35]. The VATS group showed less perioperative morbidity, but the two groups shared similar results in regional lymphadenectomy, nodal upstaging, OS, and DFS. Denlinger and colleagues’ studies discovered there was a slight trend toward more total nodes dissected in the late group as opposed to the early group [31]. Furthermore, they found no survival advantage in complete mediastinal lymph node dissection, compared to the systemic lymph node sampling in patients who underwent the surgery. Additionally, there was no survival difference between the two groups. This result, as a matter of fact, matches the American College of Surgeons Oncology Group Z0030 trial. Licht and colleagues [36] compared VATS with thoracotomy in patients receiving lobectomy for clinical stage I NSCLC. They concluded that nodal upstaging occurred significantly higher after thoracotomy. Selective reporting bias, which should and could have been avoided in the use of complete national data, is considered the main reason; nevertheless, the multivariate survival analysis remained no statistical difference in the two groups. Lichtand colleagues [36] investigated 11,513 patients from the Society of Thoracic Surgeons database undergoing anatomical pulmonary resections in clinical stage T1N0M0 or T2N0M0 lung cancer. They concluded that upstaging from N0 to N1 was more frequently observed in the thoracotomy group; however, upstaging from N0 to N2 was similar between both approaches. This finding contradicts numerous studies that have reported that open and VATS approaches result in a similar number of lymph nodes and lymph node stations being evaluated. The VATS approach was therefore chosen as the standard operation in our study, showing that lobectomy with lymph node dissection via VATS is not inferior to traditional thoracotomy. In fact, it resulted in a shorter admission period and less perioperative complications.

Regarding the long-term outcomes, the mean follow-up duration in our study was 60.6 months, which was longer than those of previous studies; however, the 5-year OS and DFS rates in our study were similar to previous studies. We extended the survival test to up to 10 years and found no statistically significant difference. We also analyzed CSS to emphasize that the survival rate was similar between the two groups. For the issue of lymph node upstaging, our upstaging rate was similar to previous research, and there was no statistical difference of the pN stage in the two groups. In summary, the VATS lobectomy with L-SND is feasible for stage I NSCLC.

The present study has several limitations. First, we did not perform a randomized controlled trial. Second, there is a selection bias owing to the retrospective study design as well as the patient selection of the L-SND group. Third, the case number in the two groups was of great discrepancy. Fourth, we used a PET or CT scan for evaluating the clinical nodal stage, leading to the upstaging. Our upstaging rate was higher than previous research, and we thought it might be caused by the small sample size. Finally, there were no uniform criteria for selecting L-SND or SND. During surgery, lymph nodes that were found to be enlarged up to 1 cm, even if there was no uptake in the staging PET scan, still required removal. These cases were converted to SND from L-SND during the surgery. Our further plan is to keep collecting cases and consulting expertise for formulating the criteria of choosing L-SND or SND. A secondary analysis from other randomized controlled trial could also be a choice for a similar topic before the establishment of criteria. Once the selection criteria are established, the randomized control trial can be run and applied.

## 5. Conclusions

A VATS lobectomy with L-SND is feasible for clinical stage I NSCLC owing to the similar surgical and long-term outcomes with SND. For those patients with lymph node uptake seen on PET scans or a more advanced stage, the standard operation by lobectomy with SND is recommended.

## Figures and Tables

**Figure 1 diagnostics-13-01399-f001:**
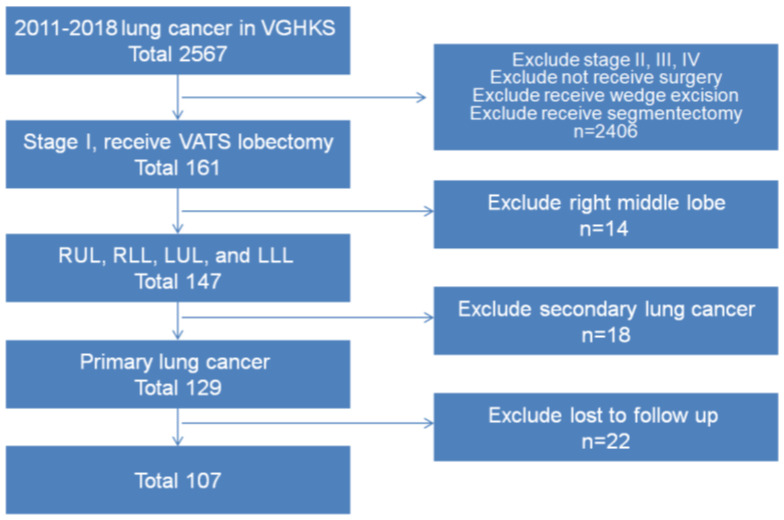
Patient selection. Inclusive and exclusive criteria.

**Figure 2 diagnostics-13-01399-f002:**
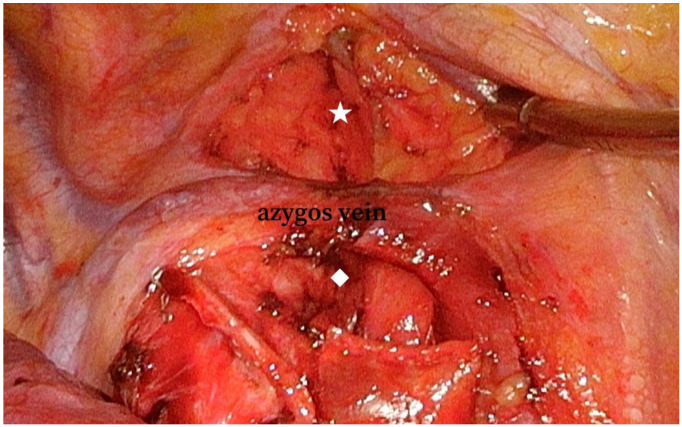
Lymph node stations 2 to 4. Asterisk: station 2; diamond: station 4; station 3 was between station 2 and 4 and the midline of the spine.

**Figure 3 diagnostics-13-01399-f003:**
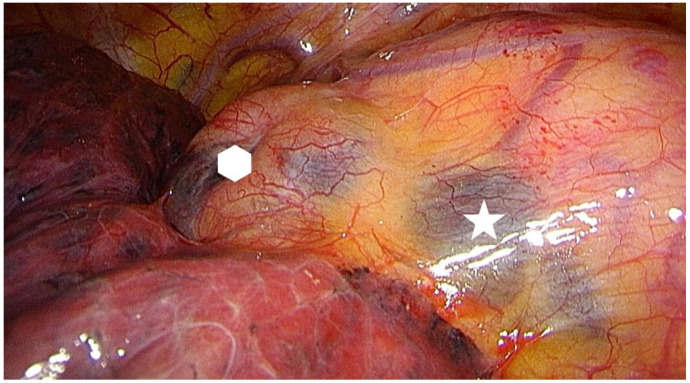
Lymph node station 5 and 6. Hexagon: station 5; asterisk: station 6.

**Figure 4 diagnostics-13-01399-f004:**
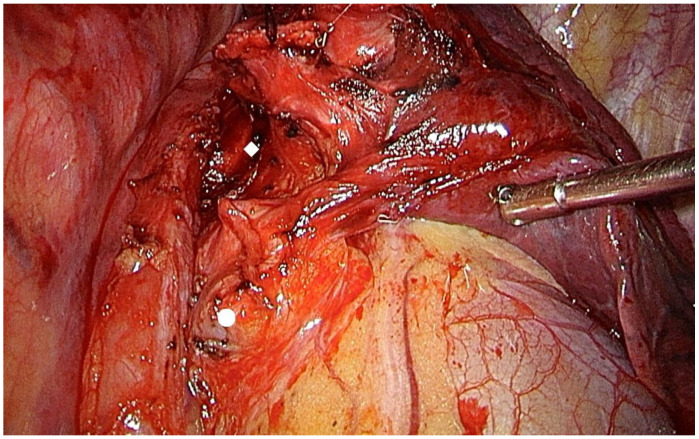
Lymph node station 7 and 8. Diamond: station 7; round: station 8.

**Figure 5 diagnostics-13-01399-f005:**
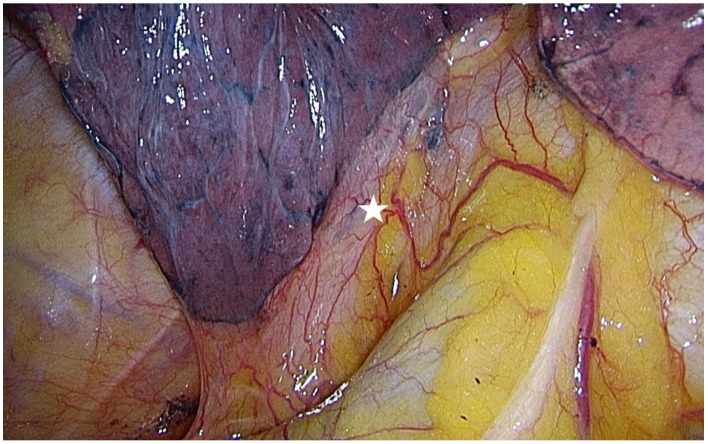
Lymph node station 9. Asterisk: station 9.

**Figure 6 diagnostics-13-01399-f006:**
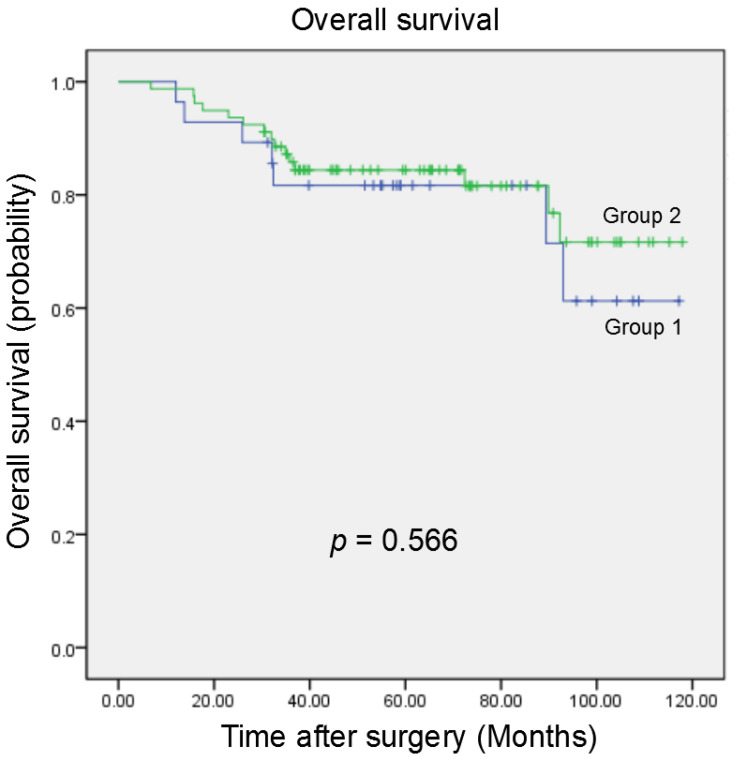
Survival outcomes: The Kaplan–Meier estimation showed no statistical difference in the OS (overall survival) rate between the lobe-specific lymph node dissection (L-SND) and systematic lymph node dissection (SND) groups.

**Figure 7 diagnostics-13-01399-f007:**
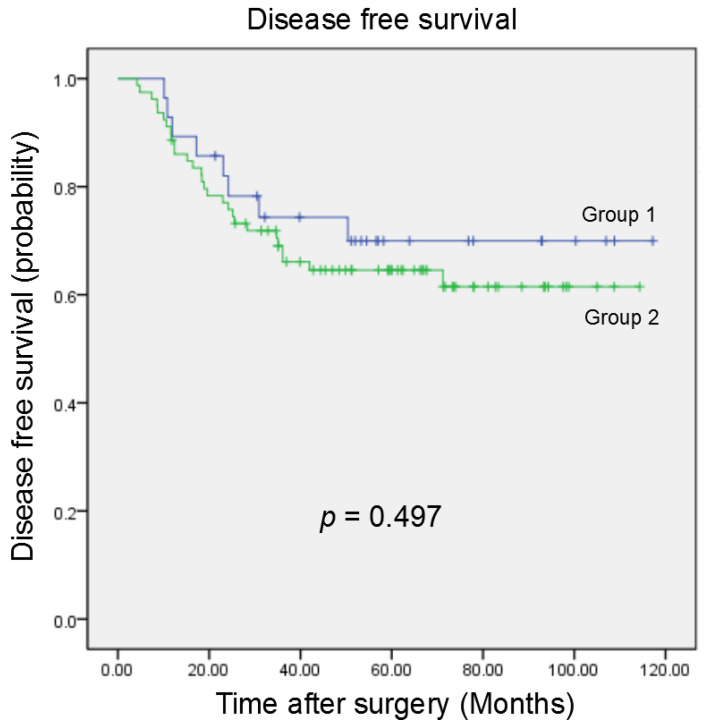
Survival outcomes: The Kaplan–Meier estimation showed no statistical difference in the disease-free survival (DFS) rate between the lobe-specific lymph node dissection (L-SND) and SND groups.

**Figure 8 diagnostics-13-01399-f008:**
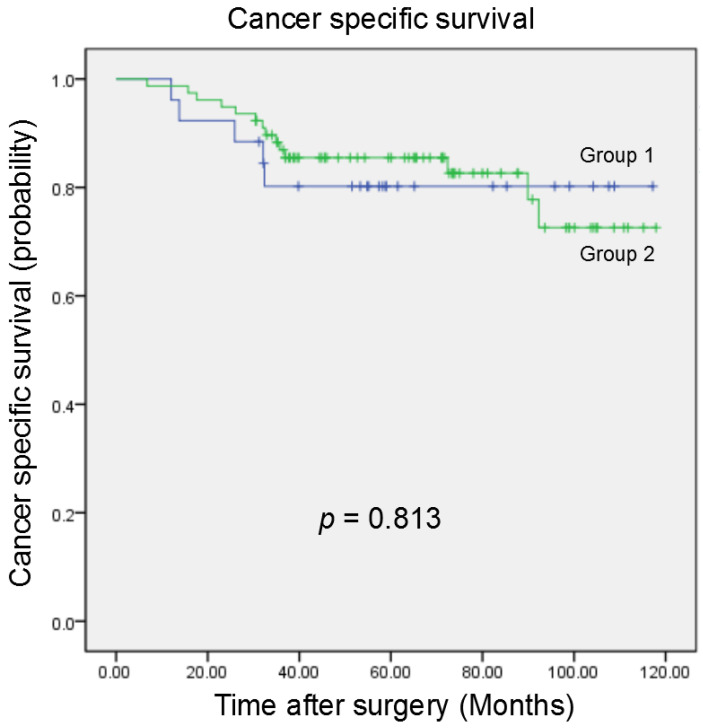
Survival outcomes: The Kaplan–Meier estimation showed no statistical difference in the cancer-specific survival (CSS) rate between the lobe-specific lymph node dissection (L-SND) and SND groups.

**Table 1 diagnostics-13-01399-t001:** Patients’ baseline characteristics.

Variables	L-SND (*n* = 28)	SND (*n* = 79)	*p*-Value
Age	61.3 ± 13.2	62.5 ± 8.4	0.643
Sex			
Male	15 (53.6%)	27 (34.2%)	0.071
Female	13 (46.4%)	52 (65.1%)	
Smoking history			
No	19 (67.9%)	58 (73.4%)	0.574
Yes	9 (32.1%)	21 (26.6%)	
Histology			
ADC	27 (96.4%)	74 (93.6%)	0.364
SCC	0	4 (5.1)	
Others	1 (3.6%)	1 (1.3%)	
Tumor location			
Right upper	7 (25%)	30 (38.0%)	0.134
Right lower	10 (35.7%)	12 (15.2%)	
Left upper	6 (21.4%)	22 (27.8%)	
Left lower	5 (17.9%)	15 (19.0%)	
Tumor size	30.0 ± 9.4	29.8 ± 8.7	0.438
Adjuvant CTx			
No	9 (32.1%)	34 (43.0%)	0.312
yes	19 (67.9%)	45 (57.0%)	
Clinical stage			
IA	10 (35.7%)	29 (36.7%)	0.925
IB	18 (64.3%)	50 (63.3%)	
Differentiate			
Well	1 (3.6%)	6 (7.6%)	0.737
Moderately	22 (78.6%)	61 (77.2%)	
Poorly	5 (17.8%)	12 (15.2%)	
pN stage			
pN0	16 (57.1%)	55 (69.6%)	0.154
pN1	8 (28.6%)	10 (12.7%)	
pN2	4 (14.3%)	14 (17.7%)	
Type 2 DM			
No	22 (78.6%)	67 (84.8%)	0.448
Yes	6 (21.4%)	12 (15.2%)	
HTN			
No	21 (75%)	48 (60.8%)	0.176
Yes	7 (25%)	31 (39.2%)	
CAD			
No	26 (92.9%)	75 (94.9%)	0.681
Yes	2 (7.1%)	4 (5.1%)	
Old CVA			
No	27 (96.4%)	75 (94.9%)	0.748
Yes	1 (3.6%)	4 (5.1%)	
FEV1/FVC (%) ^a^	81.26 ± 6.73	82.37 ± 7.09	0.886

Data are presented as number (%), or mean ± standard deviation. ^a^ Unknown for 1 patient. L-SND, lobe-specific lymph node dissection; SND, systematic lymph node dissection; ADC, adenocarcinoma; SCC, squamous cell carcinoma; CTx, chemotherapy; DM, diabetes mellitus; HTN, hypertension; CAD, coronary artery disease; CVA, cerebrovascular accident; FEV1, forced expiratory volume in 1 s; FVC, forced vital capacity.

**Table 2 diagnostics-13-01399-t002:** Surgical outcomes.

Variables	L-SND (*n* = 28)	SND (*n* = 79)	*p*-Value
Surgical time (min) ^a^	240 ± 56.7	244 ± 77.2	0.363
Admission duration (days)	8.14 ± 6.35	6.87 ± 2.61	0.312
ICU duration (days) ^b^	1.04 ± 0.20	1.06 ± 0.23	0.597

^a^ Unknown for1 patient. ^b^ Unknown for 11 patients. ICU, intensive care unit.

## Data Availability

The data presented in this study are available on request from the corresponding author.

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
