# Peer review of "Comparison of the Outcomes between Systematic Lymph Node Dissection and Lobe-Specific Lymph Node Dissection for Stage I Non-small Cell Lung Cancer"

_diagnostics, 2023, doi:10.3390/diagnostics13081399_

Round 1

Reviewer 1 Report

First, I would like to thank the Editor for giving me the opportunity to review this manuscript on a very interesting topic. Indeed, in patients affected by early-stage NSCLC, the extent of required lymph node dissection and its impact on survival is still a matter of discussion. I congratulate

The present study deals with this issue, comparing systematic lymph node dissection (SND) and lobe-specific lymph node dissection (L-SND) in clinical stage I NSCLC.  The Authors define the lobe-specific nodes as upper zone (stations 2-4 on the right side and 5-6 on the left side) and lower zone (stations 7-9).

As this is a retrospective study, the first question is: what are the precise criteria for enrolling patients in the L-SND group? It is unclear whether L-SND is a routine procedure performed in their centre or whether the patient selection was done retrospectively in order to compare standard SND with limited lymph node dissections.  If L-SND was performed routinely the second question is the following: why the station #7 was not included in all procedures, as recommended by ESTS?

The third question is: were frozen sections ever used to perform L-SND rather than SND? If the answer were affirmative, the sample would become inhomogeneous and this should be included in the exclusion criteria, in my opinion.

In conclusion, this is an interesting and well written manuscript whose scientific importance is limited by the retrospective design of the study and by the small sample size of the groups (only 28 patients in L-SND group!). On such basis this research does not allow any conclusive statement even though it could represent a small contribution to the current debate.

Author Response

Dear reviewer

We would like to thank the the Reviewers for their valuable and constructive suggestions, which have greatly improved our manuscript. The manuscript has been revised accordingly. Responses to specific comments from the reviewers are listed below.

Reviewer 2 Report

Dear Editor and Authors,

It was my pleasure to review this retrospective, single institution analysis titled “Comparison of the Outcomes between Systematic Lymph Node Dissection and Lobe-Specific Lymph Node Dissection for Stage I Non-small Cell Lung Cancer” by Dr. Huang and colleagues from the Division of Thoracic Surgery, Department of Surgery at Kaohsiung Veterans General Hospital in Kaohsiung, Taiwan.

The question the authors are trying to answer is in the forefront of thoracic surgery and one of the main points of contention in all the surgical fora, societies and meetings!! Therefore, there has been significant and much more inclusive and larger research studies on the subject. Consequently, I do not feel this study is adequate to provide meaningful and translatable results for a number of reasons. Specifically, its sample size is quite small at only 107 patients!! Considering that the group studied includes stage I patients which have a quite good prognosis following surgery the number of cases that need to be analysed to provide statistically meaningful results needs to be much higher. The authors have made no attempt to conduct a power or sample size calculation!! This is the major drawback on this study!!

The manuscript needs significant language editing because there are numerous language and expression errors that need correction. I suggest that a professional editing service or a native English speaker polish up the work!

I disagree with the statement “Furthermore, the role of lymph node dissection has been considered only for staging because there were no survival benefit between systematic mediastinal lymphadenectomy and mediastinal lymph node sampling” because as mentioned previously the judgement is still unclear on this. The authors need to re-phrase this part of their introduction.

Why did the authors only utilized in their analysis only patients which had undergone VATS lobectomy? Logically surgical technique - access is unrelated to disease free and overall survival!! Unless the authors agree that open surgery (which has much better access for lymph node dissection/sampling confers a survival benefit!!

Why were middle lobectomies excluded from the analysis?

The fact that almost half the patients had N1 or N2 disease means these patients can not be considered as stage I patients (early stage).  The authors should have used the histopathological stage and not the clinical stage in their stratification!  There needs to be statistical adjustment and a multivariable analysis to adjust for these discrepancies!! Also, it needs to be mentioned as a limitation of the study!!

There seem to be a significant percentage (39 patients) with missing data in regards to surgical outcomes. How complete were the rest of the data? Was there a significant percentage of missing data as well?

The fact that there is no significant differences in OS, DFS, CSS can most likely attributed to the small sample size of the study!! I mentioned this previously!!

Minor comments:

Figure 1 needs some minor editing – it is lost to follow up not lost of!!

Thank you for asking me to review this work.

Author Response

We would like to thank the Reviewers for their valuable and constructive suggestions, which have greatly improved our manuscript. The manuscript has been revised accordingly. Responses to specific comments from the reviewers are listed below.

Round 2

Reviewer 2 Report

Dear Editor and Authors,

Thank you for asking me to re-evaluate this work now that the authors have revised and improved it. In truth, although this is an interesting work in terms of concept and subject matter due to its small sample size and various limitations it has. However, the authors have acknowledged this in their limitation section. 

Therefore, I am now content to recomment the publication of this work with the caveat that its results need to be considered in conjunction with the limittations asserted. 

Thank you all and stay well.